# Optimization of GMAW Process Parameters in Ultra-High-Strength Steel Based on Prediction

Alnecino Netto [1], Francois Miterand Njock Bayock [2] and Paul Kah [3,*]

1   Department of Mechanical Engineering, School of Energy System, LUT-University,
    53850 Lappeenranta, Finland; alnecinonetto@gmail.com
2   Laboratory of Mechanic, Department of Mechanical Engineering, ENSET Douala, University of Douala,
    Douala P.O. Box 1872, Cameroon; njockfm1@outlook.com
3   Department of Engineering Science, University West, 461 86 Trollhättan, Sweden
*   Correspondence: paul.kah@hv.se

**Abstract:** Ultra-high-strength steel (UHSS) is a complex and sophisticated material that allows the development of products with reduced weight but increased strength and can assist, for example, in the automotive industry, saving fuel in vehicles and decreasing greenhouse gas emissions. Welding UHSS has a certain complexity, mainly due to the higher alloys and heat treatments involved, which can result in a microstructure with higher sensitivity to welding. The primary purpose of the current work was to select the best parameters of the gas metal arc welding (GMAW) for welding the S960 material based on prediction methods. To achieve the expected results, a finite element analysis (FEA) was used to simulate and evaluate the results. It was found that the welding parameters and, consequently, the heat input derived from the process greatly affected the UHSS microstructure. Using FEA and estimating the extension of the heat-affected zone (HAZ), the peak temperature, and even the effect of distortion and shrinkage was possible. With an increase in the heat input of 8.4 kJ/cm, the estimated cooling rate was around 70 °C/s. The presence of a softening area in the coarse grain heat-affected zone (CGHAZ) of welded joints was identified. These results led to an increase in the carbon content (3.4%) compared to the base metal. These results could help predict behaviors or microstructures based on a few changes in the welding parameters.

**Keywords:** ultra-high-strength steel; gas metal arc welding; heat input; welding simulation; microstructural constituents

## 1. Introduction

The first indication of a possible oil emergency occurred in 1975. Thus, the necessity of reducing fuel consumption resulted in new aims for automotive companies and a need to decrease their vehicles' weights. Over the years, fuel consumption reduction goals have become more intensive and have reached different areas, changing the methods of designing new products by thinking about how to prevent excessive weight and keep the desired properties of the materials. Additionally, due to environmental issues, there is also the necessity to decrease gas emissions generated by fuel consumption and reduce the energy utilized to produce materials to ensure a more sustainable environment [1–3]. In this way, advanced-high-strength steel (AHSS) and ultra-high-strength steel (UHSS) have come to mitigate these problems. Another exciting perspective in regard to UHSS or very-high-strength steel (VHSS), as demonstrated by Qiang et al. [4], is related to the benefits of logistics and operations. Since building a structure takes time and a certain number of materials, using materials with higher strengths could decrease transportation and reduce other processes, such as the amount of welding required.

First, defining certain terms used in the current work is essential. Depending on the author or sources, AHSS and UHSS designations can vary from the definitions used in the present research. However, the present study considers the exact designation of UHSS and

AHSS as all steel with an ultimate tensile and yield strength of above 700 MPa. AHSS was developed over many years, today enabling the production of steel sheets, which has many benefits compared to other lightweight materials. However, according to Fonstein et al. [2], Tumer [5], and Demeri et al. [6], to reach the current high-strength steel level with a range of 500 to 1700 MPa and improved ductility, it is necessary to consider different methods during its production. Thus, changes in the microstructure through steel-strengthening approaches, the addition of alloys and micro-alloy elements, and heat treatments are the methods most commonly used to achieve excellent results of AHSS [7–10]. In this way, many grades and classifications are created, allowing combinations for many applications.

Currently, AHSS is in its third generation. Many studies that involve welding effects have been developed along with the evolution of new grades of AHSS with critical mechanical properties and a focus on essential properties, such as strength and toughness [11–13]. Relating to the properties of UHSS, Neimitz et al. [14] demonstrated that the characteristics of 960 QC steel are unusual compared to those of conventional ferritic steels. In this way, they undertook a study to understand whether the material's behavior was unsuitable for a typical material–master curve. As an example of this result, it was observed that there were no differences between plate thicknesses, ranging between 4 and 8 mm, in relation to fracture toughness.

The development of UHSS and its applications has brought different perspectives to conventional production methods, primarily due to the unique characteristics of the product/structures to which UHSS is applied. These characteristics may create not only advantages but also difficulties that need to be discussed before these new materials are widespread in order to prevent future barriers [11–15]. According to Jenney et al. [16], in certain manufacturing processes (including welding), the improvement of materials plays an essential role, mainly affecting properties, such as hardness and weldability, that need to be enhanced. Accordingly, the chemical composition and microstructure of steel are some of the main issues that need to be considered when the material is modified.

The development of new grades of AHSS and UHSS with different characteristics is being studied and evaluated to enable their utilization in welding processes. In a study related to the ultra-narrow gap laser and gas metal arc welding (GMAW) process outlined by Guo et al. [10], the S960 (structural steel with 960 MPa as minimum yield strength) developed by Tata Steels was analyzed. The S960 is a steel that has excellent benefits, such as good impact toughness, weldability, and low weight, in addition to its high strength, which makes this material a good option for applications in heavy cranes, oil and gas transportation pipes, offshore industries, and shipbuilding [17–20]. The S960 is a high-strength low-alloy (HSLA), where arc welding processes are typically used; however, the softening of the heat-affected zone is a typical issue in this welding process. The thermo-mechanical material properties of S960 using the GMAW process have also been analyzed in other studies by Bhatti et al. [13], Schaupp et al. [19], and Bayock et al. [20].

In a mechanical property evaluation of dual-phase (DP) steel through gas metal arc welding (GMAW), Ramazani et al. [9] also mentioned the influence of the heat input in the welding process on the microstructure of the materials. The increase in the heat input and the temperature gradient causes changes to the mechanical properties of DP steel. Furthermore, relevant changes in the material's microstructure can occur due to the higher heat input of the GMAW process and lower cooling rates.

The selection of AHSS/UHSS in welding applications requires a detailed evaluation that demonstrates both the benefits and disadvantages. As previously mentioned, the ability to reduce the material's volume without compromising the safety and quality of welded structures is an essential factor in the selection of AHSS/UHSS. Thus, the welding parameters may need to be optimized to guarantee a reliable product. Along with optimization, it is fundamental to consider computational tools that can reduce the development time of a product [21–24]. However, while most simulation software available today performs different functions, only a few are designed specifically for welding simulations. The multi-task simulation software offers excellent results, but the

time required to model a specific welding operation must be optimized [25–28]. Therefore, the necessity to obtain faster and cheaper results and, consequently, high-quality products can constantly stimulate the development of specialized and user-friendly simulation software in the welding process.

The current research focuses on some of the main problems in welding UHSS materials based on the proper choice of welding parameters and the influence of the heat input to achieve high quality and, consequently, a reliable weld. The questions raised on welding UHSS are mostly related to the sensitivity of the material due to a high number of alloy elements. Currently, there is a wide range of AHSS and UHSS. However, due to the differences in welding conditions and the chemical composition of each grade, the same welding parameters may only apply to some grades of the material. Additionally, the different methods of heat treatment used to ensure the properties of each grade of the UHSS may also provide difficulties in the welding process.

In this way, the constant development of AHSS is also a motivation for performing research on this material since there needs to be more information in GMAW using the UHSS S960 material during the development of the current thesis. Finally, optimizing the UHSS welded structures by predicting the welding parameters in GMAW is an alternative for obtaining better quality welds, preventing possible failures, and contributing to the continuous use and development of AHSS and UHSS.

This research investigated a UHSS S960 material welded by the GMAW process through a welding simulation using finite element analysis. Due to the limited research on the UHSS S960 conducted to date, the effects of the heat input, microstructure, and mechanical properties in GMAW were analyzed. A three-dimensional (3D) model and weld modeling were generated during the welding simulation. To validate the finite element model, experimental results were used as an input, and a comparison between the virtual and physical models was accomplished. The effects of the heat input in the microstructure were examined by single and multiple passes to evaluate the results and adequately optimize the welding process parameters.

## 2. Materials and Methods

### 2.1. Materials Properties

During the experimental tests, a base and a filler material were used for welding. The base material was the UHSS Optim® 960 QC (quenched and cold formable) with a 5 mm thick plate. It was provided by the Ruukki Company, followed by a material certificate with the necessary information related to the metal.

It was essential to calculate the carbon equivalent (CE) to understand the metallurgy and weldability effects during welding. For example, a 900 MPa yield strength steel grade generally has worse weldability than the same type of steel with a 700 MPa yield strength. This occurs due to a microstructure that is characteristic of a material with a higher yield strength and a higher carbon equivalent with increased hardenability, i.e., a sensitive microstructure [6,29]. A study carried out by Martis et al. [30] showed difficulty during welding in steel with a high carbon content (around 0.4%) and a consequently higher carbon equivalent. Considering this, the recommended carbon content should be below 0.1%, which is preferably prioritized if compared to a high carbon equivalent [6,31]. However, due to the production cost, the first alloying component usually used to increase the strength of steel is carbon [32–36].

The CE (carbon equivalent) obtained in the present research was calculated based on the following equation recommended by the International Institute of Welding (IIW) [31,37]:

$$CE = \%C + \frac{\%M_n}{6} + \frac{\%C_r + \%M_o + \%V}{5} + \frac{\%N_i + \%C_u}{15} \qquad (1)$$

In Equation (1), CE is the carbon equivalent, C is carbon, Mn is manganese, Cr is chromium, Mo is molybdenum, V is vanadium, Ni is nickel, and Cu is copper. Thus, with the chemical composition obtained by the material certificate of the base material, it was

possible to obtain a 0.47 CE (Table 1). Based on the CE result, the weldability of the S960 was good, and no preheating was needed to accomplish the welding of a 5 mm thickness plate.

**Table 1.** Chemical composition and carbon equivalent of the base material S960 steel (wt. %) [20].

| C | Si | Mn | P | S | Al | Cr | Mo | Ti | B | Nb | Ni | V | Cu | Fe | CE |
|---|----|----|---|---|----|----|----|----|---|----|----|---|----|----|----|
| **0.09** | 0.21 | 1.05 | 0.01 | 0.004 | 0.03 | 0.82 | 0.158 | 0.032 | 0.0019 | 0.003 | 0.04 | 0.008 | 0.025 | balance | 0.47 |

The mechanical properties of the UHSS 960QC are shown in Table 2. The same table also presents information related to the nominal and measured values. In the case of the measured condition, the values presented were obtained from the information provided in the material certificate.

**Table 2.** Mechanical properties of the base metal UHSS S960 [3,6,20].

| Result | Yield Strength (Mpa) Min. | Tensile Strength (Mpa) Min. | Elongation (%) Min. |
|--------|---------------------------|-----------------------------|---------------------|
| Nominal | 960 | 1000 | 7 |
| Measured | 976 | 1108 | 12 |

The filler wire material used in this study was the Union X96 (ER120S-G) [18]: a solid wire with a diameter of 1.0 mm. The chemical composition and the respective CE of the filler material are shown in Tables 3 and 4, which present the mechanical properties.

**Table 3.** Chemical composition and carbon equivalent of filler wire Union X 96 (wt. %) [3,6,20].

| C | Si | Mn | Cr | Mo | Ni | Fe | CE |
|---|----|----|----|----|----|----|----|
| 0.12 | 0.80 | 1.90 | 0.45 | 0.55 | 2.35 | Balance | 0.79 |

**Table 4.** Mechanical properties of the filler wire Union X96 [20].

| Yield Strength (Mpa). | Tensile Strength (Mpa) | Elongation (%) |
|-----------------------|------------------------|----------------|
| 930 | 980 | 14 |

### 2.2. Welding Setup

To accomplish the experimental tests, the base metal was fixed in a workbench through fixtures and was connected to five thermocouple probes (Figure 1). Utilizing thermocouple probes allowed us to measure the temperature in a region of the welding during a specific time. In this way, it was possible to obtain values in relation to the peak temperature to support an understanding of the changes in the microstructure of the weld.

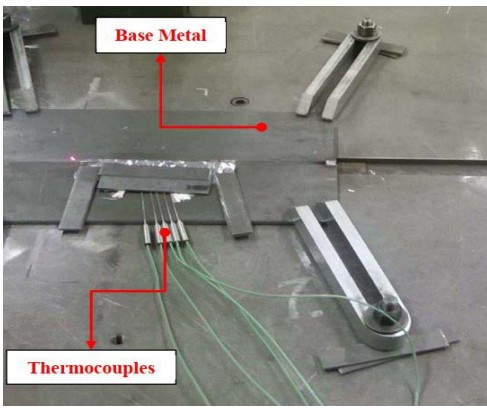

**Figure 1.** Thermocouple and base metal position.

The thermocouple probes were positioned in different coordinates of the base material to evaluate the thermal cycles and microstructure changes based on the temperature variation in different welding regions. Figure 2 shows the five drilled holes to position the thermocouples and the X and Y axis. The coordinate position of each thermocouple probe is shown in millimeters in Table 5.

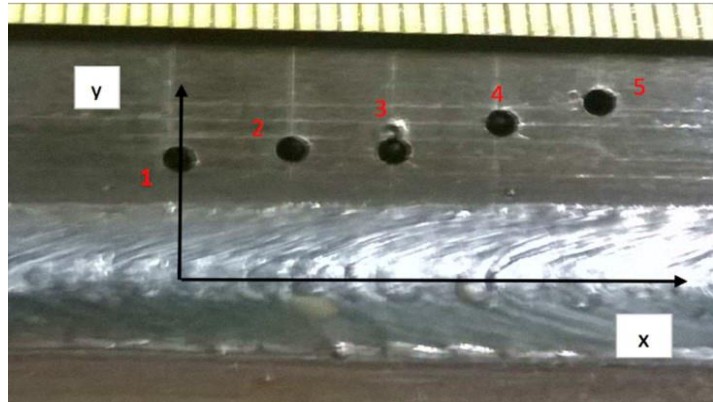

**Figure 2.** Location of the five (5) thermocouple probes in the experimental analysis.

**Table 5.** Positioning coordinate of the thermocouple probe (in millimeters).

| Probe | Axis X | Axis Y |
|-------|--------|--------|
| 1 | 0 | 6.8 |
| 2 | 5 | 7.5 |
| 3 | 10 | 7.4 |
| 4 | 15 | 9.5 |
| 5 | 20 | 10.5 |

As previously mentioned, fixtures were used to hold the plates in the correct position maintaining the stability of the pieces during the welding process. Furthermore, a welding robot was used to perform the welding (Figure 3). The welding robot manufactured by the ABB Group used in the present study was fully able to weld by GMAW [38–40].

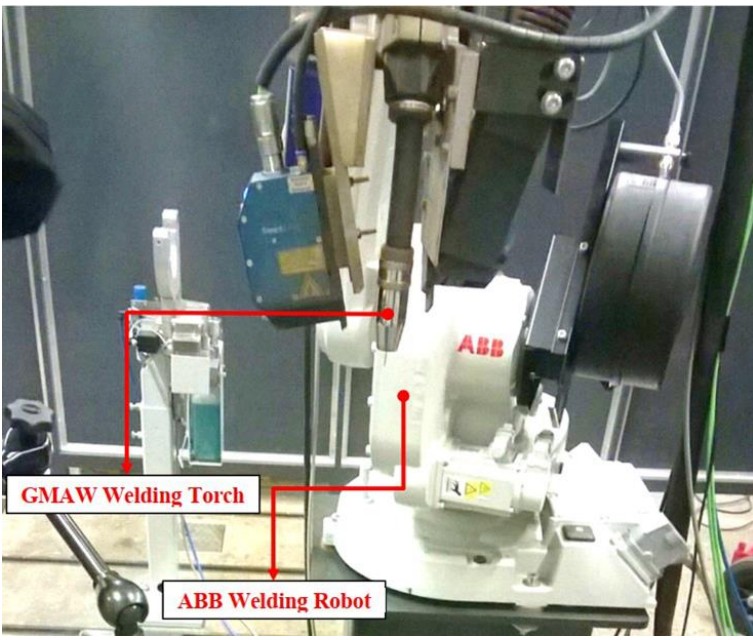

**Figure 3.** ABB welding robot utilized during the experiment.

From Figure 3, it is also possible to visualize the robot's manipulator settled with a torch in the GMAW process. Using a robot provided better consistency during the welding, mainly due to the advantage of keeping the same velocity and position and avoiding excessive variations along the process. The welding process performed by a robot could also reduce variation in the penetration during welding.

### 2.3. Welding Parameters

In the present research, the welding joint selected was a butt joint with a square groove weld-type configuration. This type of welding joint was chosen due to the base metal's lower thickness (5 mm). Moreover, the welding parameters and conditions were defined based on the selected welding process and joint. The shielding gas used to protect the welding was a mixture of 90% argon with 10% $CO_2$ with a flow rate of 16 L/min. The welding position was flat (PA) with a stick-out length of 18 mm, and the wire feed rate was 9 m/min. In relation to the air gap, it is essential to mention that the gap varied along the groove from 0 mm to 0.5 mm, mainly due to the usual deformations on the base material.

Other essential parameters that were utilized during the experimental test are shown in Table 6. This table describes the welding pass, voltage, current, welding speed, efficiency, and heat input. Due to the 5 mm thickness of the plate, only one (1) welding pass was considered. The efficiency of the GMAW process examined in this experiment was defined as 80%.

**Table 6.** Welding parameters for the experimental test.

| Welding Passes | Voltage (V) | Current (A) | Welding Speed (mm/s) | Efficiency (%) | Heat Input (kJ/mm) |
|---|---|---|---|---|---|
| 1 | 24 | 180 | 7 | 80 | 0.49 |

### 2.4. Hardness and SEM/EDS Microstructure Equipment

A hardness test was conducted for a line of measurement using a Wilson Wolpert 452SVD Vickers hardness tester (ITW, Chicago, IL, USA) according to ISO 6507-1:2018. The measurements were taken point-by-point over the entire length of the sample.

Figure 4 shows the scanning electron microscopy (SEM) and energy-dispersive X-ray spectroscopy (EDS) device (Hitachi SU3500, America, Chicago, IL, USA). EDS identified the variation in the alloy element compositions in the WM, CGHAZ, and FGHAZ. To carry out these tests, they first had to be dipped in an acetone solution to remove the accumulated carbon layer from the sample surface. The liquid had to be removed with a $CO_2$-skimming gun to obtain a dry surface. After that, the sample was placed inside the SEM/EDS device for checking with different magnifications.

During the experimental test, the main parameters, such as voltage, current, and welding speed, were defined based on the experience of the researchers in the welding laboratory. However, further in this study, some different values of these variables were considered. The heat input value was essential because it directly related to the welding parameters and could influence their optimization. To obtain the value of the heat input, some welding parameters were considered. The heat input for arc welding could be calculated by the following equation [13,18]:

$$Q = \frac{(E \times I) \times \eta}{v} \tag{2}$$

In Equation (2), $Q$ represents the heat input (J/mm), $E$ is the voltage, $I$ is the current, $\eta$ is the arc efficiency, and $v$ is the welding speed (mm/s). The influence of heat input and the welding wire greatly influenced the GMAW process. In addition, Mohrbacher, H. [8] mentioned that to achieve minimum nominal tensile strength results for a 700 Mpa grade steel, the heat input must be restricted to a maximum of 11 kJ/cm (1.1 kJ/mm).

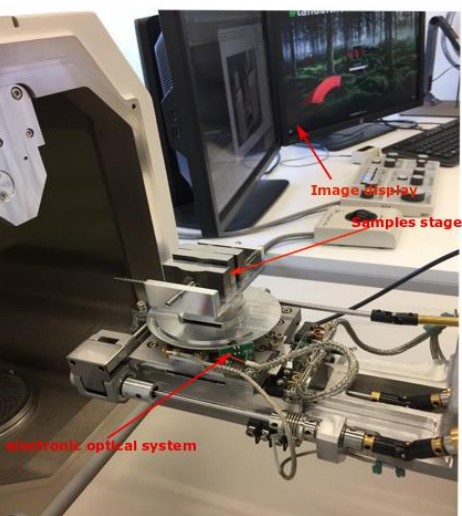

**Figure 4.** SEM/EDS test equipment.

*2.5. Welding Simulation Development*

During the welding simulation, a virtual model was analyzed by finite element analysis to compare the results found in the experimental test with the physical model. The academic version of the software ANSYS R15.0 was used to perform the FEA. To reduce the computational processing time, the size of the specimen was purposely small. However, with satisfactory results acquired by the reduced-size model, further studies to simulate larger models are possible. In the virtual model, the same 5 mm thickness was used and considered in the physical model by single- and multi-pass welding on the S960 UHSS material.

Similarly, a butt joint with a square groove configuration was also used for the welding simulation. The groove configuration was selected based on the thickness of the material along the square groove, which was the same as in the physical model. The groove weld utilized during the simulations and the dimensions of the virtual plate are detailed in Figure 5.

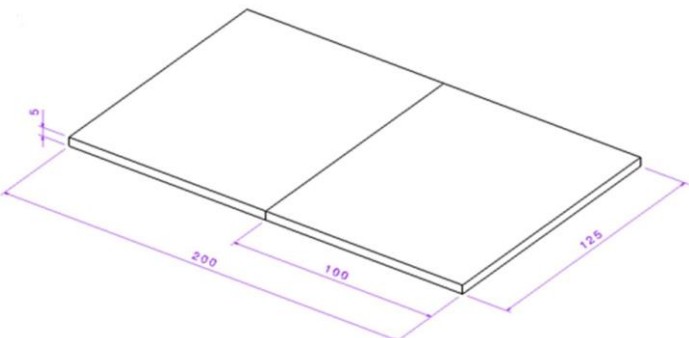

**Figure 5.** Geometric dimensions of the weld plate considered in the welding simulation.

Along with the geometric dimensions of the model, it was also necessary to generate a mesh in the virtual model. The mesh in a simulation by FEA is a technique that divides the whole model into small elements to accomplish the necessary calculation efficiently. Therefore, a refined mesh was created in all the models. The total number of elements was 2.220, and the total number of nodes was 16.216. Figure 6 shows the virtual model after the creation of the mesh.

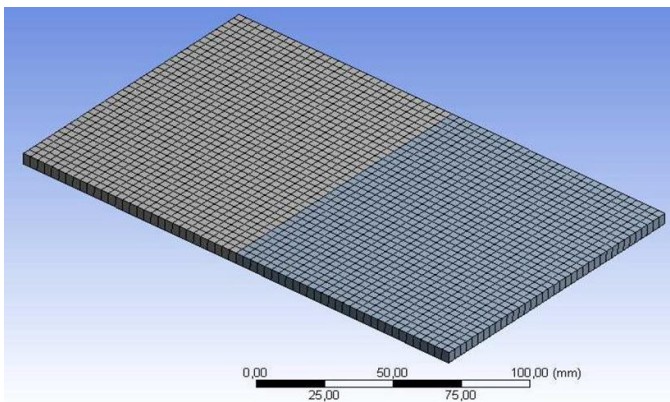

**Figure 6.** Mesh created on the finite element model.

To compare the results between the physical and the virtual models, five (5) thermocouple probes were designed in the finite element model. The position of the probes remained the same as those used in the physical model (Figure 7). In this way, using thermocouples allowed us to understand the amount of heat that was input by single- and multi-pass welding.

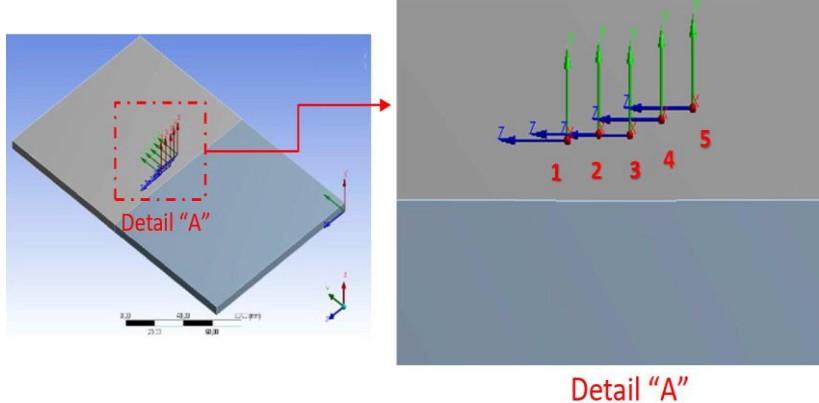

**Figure 7.** Location of five (5) thermocouple probes in the finite element model.

In the welding, the simulation used two different amounts of heat input into each welding pass. Table 7 shows the welding parameters used during the analysis. However, as the finite element analysis only considered the welding speed and the heat input, the other variables were filled to reach the desired heat input.

**Table 7.** Welding parameters used during the welding simulation by finite element.

| Welding Passes | Voltage (V) | Current (A) | Welding Speed (mm/s) | Efficiency (%) | Heat Input (kJ/mm) |
|---|---|---|---|---|---|
| 1 | 24 | 180 | 7 | 80 | 0.49 |
| 2 | 30 | 290 | 8.3 | 80 | 0.84 |

Therefore, two different heat input ranges were selected: 0.49 kJ/mm and 0.84 kJ/mm. Since the experiment of the physical model used 0.49 kJ/mm, the virtual model was considered as a higher amount (0.84 kJ/mm). In this way, the influence of heat could be compared using the same 5 mm thickness of the evaluated material.

In addition, it is essential to mention that the finite element analysis was also responsible for evaluating the thermo-mechanical behavior of the virtual models. Initially, the simulation of the welding process considers the transient thermal properties, and after that, it considers the obtained information as an input to the thermal stress analysis.

Thus, the present study provides information in relation to the distortions and stresses to analyze the influence of the amount of heat input. To obtain these results, ANSYS software was based on the Gaussian heat source, wherein the moving heat flux was calculated from the following equation [31]:

$$the\ q = C_2 e^{\frac{(x-x_0)^2 + (y-y_0)^2 + (z-z_0)^2}{C_1^2}} \tag{3}$$

In Equation (3), $q$ is the heat flux, $C_1$ is the radius of the beam, $C_2$ is the source power intensity, and $x$, $xo$, $y$, $yo$, $z$, and $zo$ are the position of the center of the heat flux in a determined route at the distance of the velocity ($v$) of the moving heat source multiplied by the time ($t$) from the start point.

## 3. Results

### 3.1. Experimental Tests' Results

To optimize the GMAW process parameters, the influence of the heat input is essential to understanding the effects of each isolated parameter. In addition to the increase in quality and reliability in a structure, the right choice of welding parameters was also important to avoid the waste of unnecessary energy to accomplish welding.

The experimental test was carried out in the welding laboratory of the Lappeen-ranta University of Technology. To complete an efficient experimental test, some essential variables were evaluated to better comprehend the performance of the GMAW process parameters along the UHSS material. In this way, a macro-structure, micro-hardness, and an evaluation of the thermal cycle were accomplished.

A macro etch cross-section on a welding joint could be accomplished to evaluate welding structures and visually obtain results in relation to the weld penetration, dilution, weld leg dimensions, and other characteristics that might be evidenced only by a macro evaluation. In this manner, a macro etch cross-section was accomplished on the physical sample and welded by a single pass. The sample specimen was etched at the initial solution of a 4% solution of $HNO_3$ in ethanol [39].

As shown in the macro-structure result (Figure 8) of the physical experiment, it was possible to see the characteristic "V-shape" of GMAW welding, full penetration, and no defects such as cracks, porosity, inclusion, undercut, or incomplete fusion. In addition, it could be seen that the different regions affected by the heat of the welding thermal cycles were visually eased by the bounds inserted in the image. Thus, the fusion zone (FZ) was around 4 mm on the top and decreased to approximately 2 mm on the bottom of the joint. The heat-affected zone was around 3 mm and was divided by coarse grain (CGHAZ), which was located adjacent to the fusion zone, and fine grain (FGHAZ) was adjacent to the base material (BM).

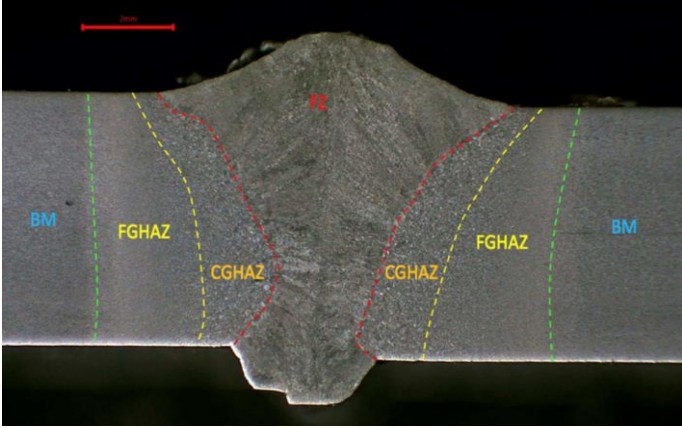

**Figure 8.** Macro section sample showing the fusion zone (FZ), coarse grain HAZ (CGHAZ), fine grain HAZ (FGHAZ), and base metal (BM).

It is also important to mention that the macro etch cross-section did not represent exactly a picture of the whole welding; however, it aided in obtaining an overview of the general welding condition. Thus, based on the information of a visual condition as shown in Figure 7, it was possible to assume that the welding parameters considered on the analyzed material and welding process were appropriate.

### 3.2. Micro-Hardness

The hard results of the physical experiment are presented in Figure 9, where it was also possible to visualize each region where the hardness test machine penetrated the material (around 0.5 mm of distance from each point) [37]. The results present a distinction in the hardness result between the BM, HAZ, and FZ. In the BM, the average hardness was around 350 HV (hardness Vickers); however, it could be seen that when approximate to the HAZ, the values of hardness tended to be lower (250 to 300 HV), which is characteristic of the region softened by lower heat input. The only exception of the lower hardness on the HAZ was the peak hardness of around 340 HV, which was found exactly between the FGHAZ and CGHAZ.

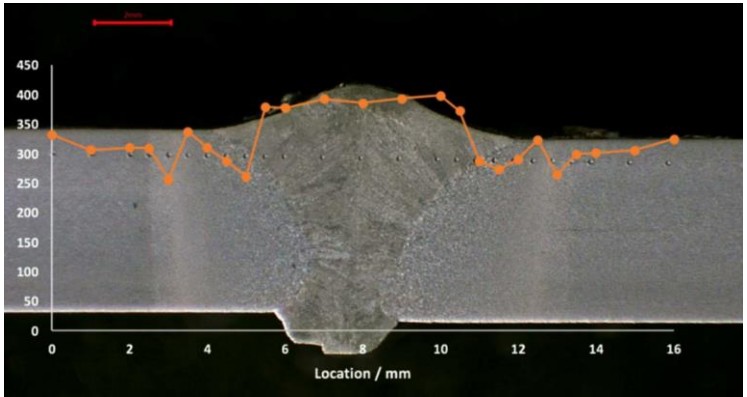

**Figure 9.** Hardness results (HV5) of the experimental test.

On the other hand, the FZ showed a higher hardness value (around 400 HV), mostly due to the higher heat input in this region and the influence of the filler material. If we analyze some defects that could occur on a welding joint in relation to the hardness, cold cracking is an example and an alternative when mitigating this menace to control the peak hardness value on the HAZ. In this way, in steel with low carbon content (around 0.06%), the peak hardness could remain below 350 HV [6,38,39].

As mentioned before, the chemical composition of the material used was relevant to understanding the hardness results. Thus, it was essential to evaluate early on the carbon content of the material to predict some effects from the welding heat input. O'Brien, A. [15] observed that the HAZ hardness is commonly used to assess the cracking susceptibility that is also correlated to the microstructure of the HAZ. The volume fraction of martensite might be a better indicator of cracking susceptibility. HAZ hardness is a good indicator of the martensite volume for similar chemical compositions; however, the hardness of martensite can vary with carbon levels. Lower carbon materials tend to crack at lower hardness levels. For in-service applications, trade-offs could be made between the HAZ hardness, hydrogen level, and chemical composition. A HAZ hardness of 350 HV might be too conservative for some applications and not conservative enough for others. Since hardness has a direct relationship with the microstructure, it is important to mention a study accomplished by Singh et al. [29], stating that both retained austenite and transformed martensite sizes need to be considered regarding the mechanical properties. Thus, with a higher martensite content, the strength also increased, and the ductility decreased. In addition, it was not only the retained austenite content that influenced the ductility but also the increase in the Mn content.

As a complement of the results already obtained by the hardness test result, through the analysis from the welding laboratory, the continuous cooling transformation (CCT) diagram of unprocessed Optim® 960 QC (Figure 10) presented much relevant information on the base material under a certain temperature. Thus, it was possible to predict the microstructure phases of the material through cooling rate information and relate it to hardness. Regarding the micro-structure, it is possible to visualize that when the hardness value was reached, a bainite structure was formed around a cooling rate of 40 °C/s suitable to the HAZ of the physical sample. However, to reach higher values of around 400 HV on the FZ, the micro-structure-formed martensite and bainite structure would need to be related to a cooling rate of around 70 °C/s.

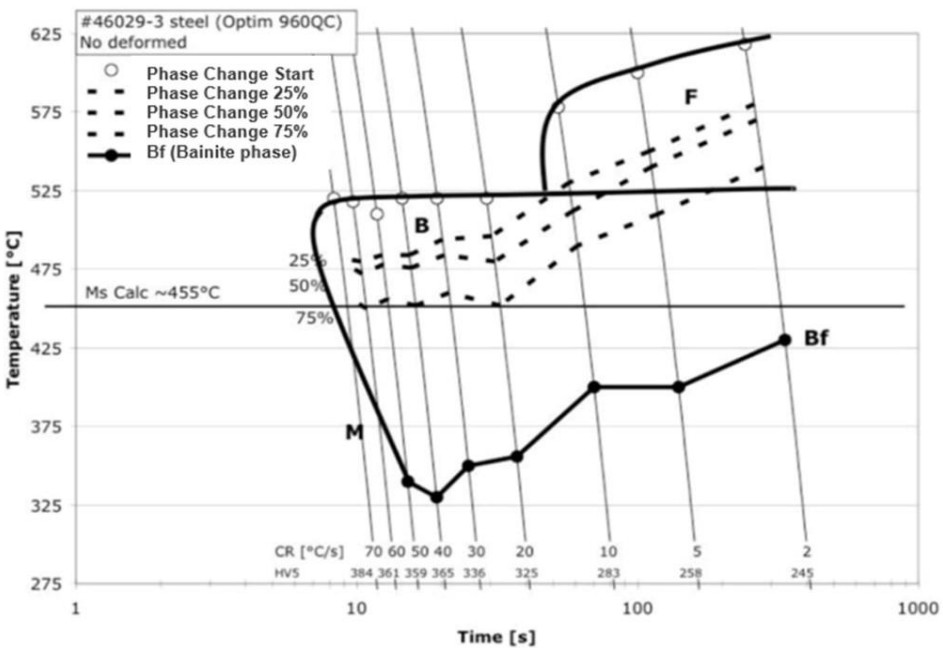

**Figure 10.** CCT diagram for unprocessed Optim® 960 QC material.

The results found on the CCT diagram were directly influenced by the heat input, given that it was derived from the welding parameters. A similar material was evaluated by Mohrbacher, H. [8] and Bayock et al. [20], and a limit of 1.2 kJ/mm was considered as the maximum heat input on the GMAW process to avoid softening in the HAZ. On the contrary, when a value below 1.0 kJ/mm was considered, unwanted hardness peaks could appear in which the HAZ became susceptible to cold cracking.

*3.3. Microstructural Constituent S960 QC Welded Joints*

Figure 11 shows an SEM micrograph of the different areas of the heat-affected zone. In the HAZ, the weld metal represented by Figure 11a, the CGHAZ (Figure 11b), and FGHAZ can be seen (Figure 11c). The typical microstructure of S960QC was mostly the formation of fine bainite (B) and Martensite (M). As the CCT diagram (Figure 10 above) observes, the welded joint microstructural analysis had input values of Q = 0.84 kJ/mm, v = 5 mm/s. The estimated cooling rate was around 70 °C/s. EDS analysis was calibrated at a voltage of 15 kV with mapping at 20 kV, and the image resolution was 1024 by 768. In the weld metal (Figure 11a), the results revealed mainly the presence of acicular ferrite (AF), a consequence of the amount of pro-eutectoid ferrite (PF) at the temperature of around 600 °C. The EDS mapping point revealed an increase in the carbon content at 3.3%. Compared to the initial alloy element composition of the filler wire, decreases in Ni (0.4%), Mn (1.2%), Cr (0.2%), and Si (0.4%) were noted. From Figure 11b, it can be seen that the microstructural constituent in the CGHAZ was a formation of tempered martensite (TMA), bainite (B), and some traces of retained austenite. EDS mapping revealed the production

of 3.4% carbon content, 0.30% of Si, 0.10% of Cr, and 1.80% of Mn. Compared to the BM, there was a decrease in Cr and Mn. The absence of Mo and Ni was also noticed, which could cause a softening in the CGHAZ. This softening area is illustrated in Figure 9 when microhardness was analyzed. The little increase in Mn promoted martensite's presence at the transformation's end. The same results were observed in FGHAZ. The alloy element composition in that zone caused a continuous decrease in Cr (0.1%), Si (0.3%), and the absence of Ni and Si. It is important to notice the fact that these alloy elements (Mo, Ni, Mn, and Cr) played an important role in the correlation between microstructural constituents and mechanical behavior when ultra-high-strength steel was wielded.

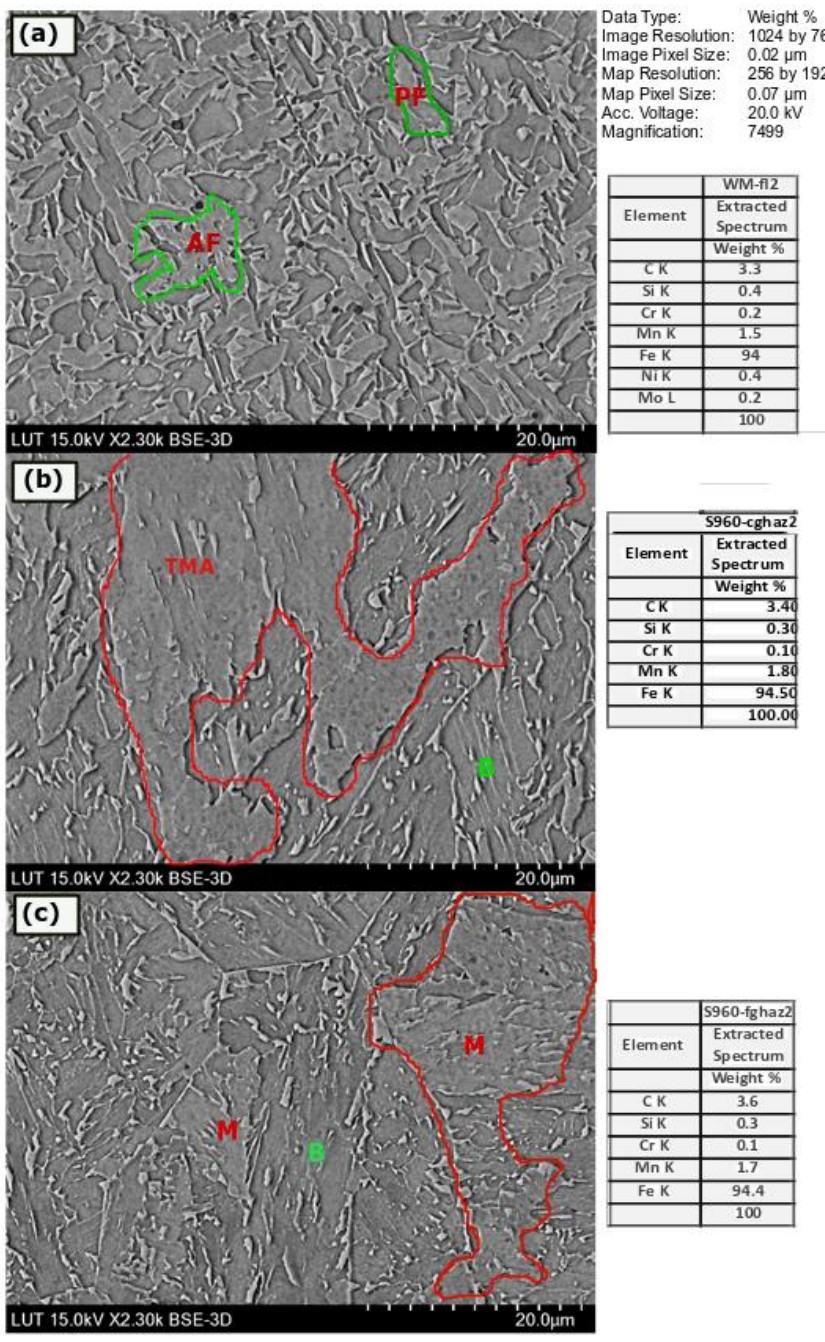

**Figure 11.** SEM images and EDS analysis formed in the HAZ of the S960QC welded joint. (**a**) weld metal microstructure showing the presence of acicular ferrite (AF) and pro-eutectoid ferrite (PF), (**b**) coarse grain heat affected zone with (TMA), bainite (B), and (**c**) fine grain heat affected zone microstructure showing formation of fine bainite (B) and Martensite (M).

### 3.4. Thermal Cycle by Thermocouples and Simulation Results

As presented before the experimental procedure, five thermocouples located on the top surface of the plates were used to measure the behavior of the temperature during the welding. Thus, the measured peak temperature captured by the thermocouple with a distance of 6.8 mm (closest to the centerline welding) was approximately 42 °C/s, which was well-suited to the microhardness result found if the data of Figure 12 are considered. It must be remembered that in FZ, a thermocouple to evaluate the 70 °C/s cooling rate was presented before it was adequate for the CCT of the material. The cooling rate value at 500 °C (R500) was based on the following equation [18,20]:

$$R_{500} = \frac{\partial T(0,T)}{\partial T} = -2\pi\lambda c\rho \frac{(T_{max} - T_0)^3}{\left[\frac{q}{(vh)}\right]^2} \tag{4}$$

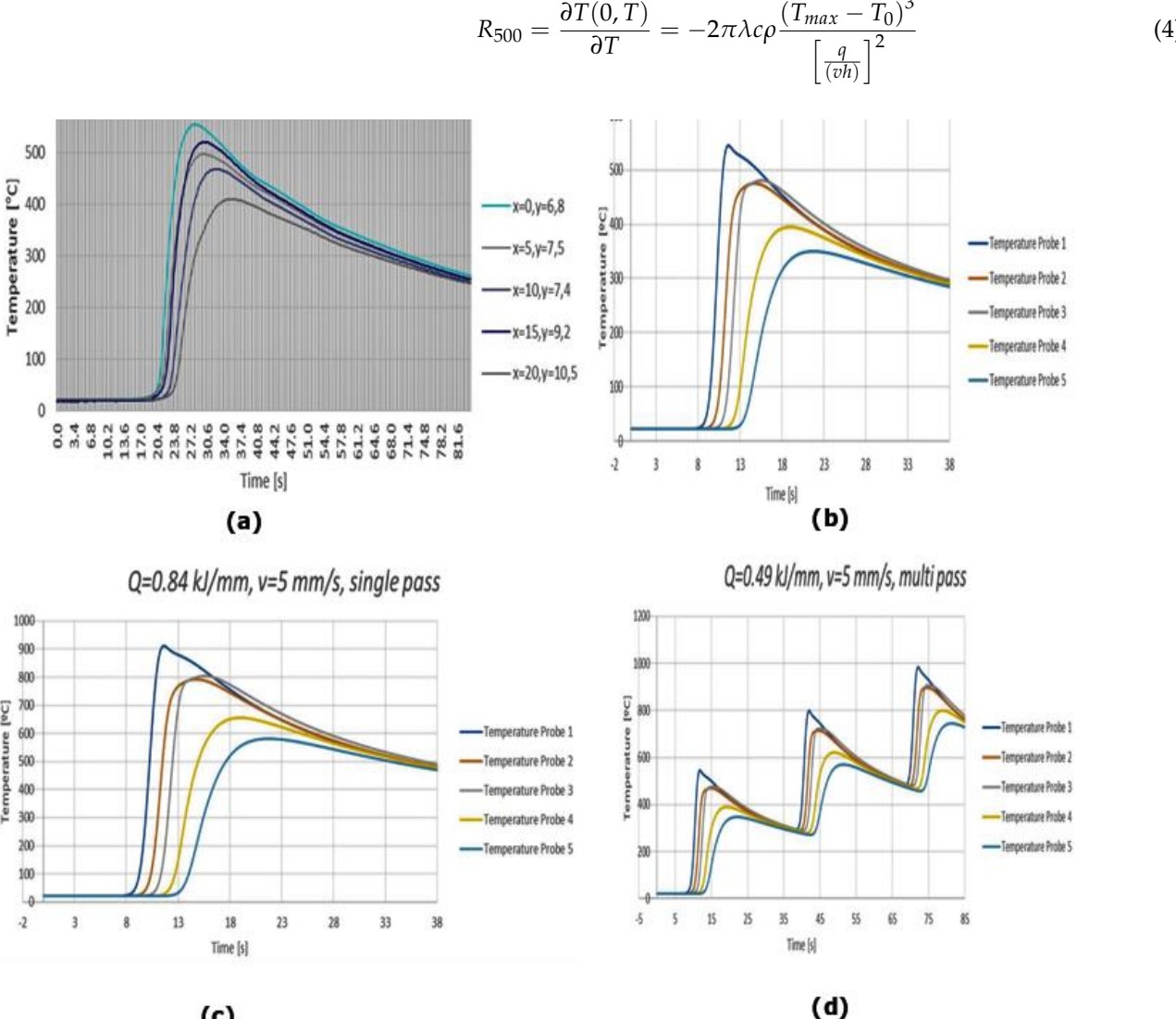

**Figure 12.** Experimental and welding simulation results of thermo cycle including (**a**) thermo cycle obtained by physical sample experiment, (**b**) simulation of single-pass welding at Q = 0.49 kJ/mm, v = 5 mm/s, (**c**) simulation of single-pass welding at Q = 0.84 kJ/mm, v = 5 mm/s, and (**d**) simulation of multi-pass welding at Q = 0.49 kJ/mm, v = 5 mm/s.

In Equation (4), $T_{max} - T_0$ was defined as the increase in the temperature during welding and $\frac{q}{(v)}$ as the heat density. $\lambda$ is the thermal conductivity, and $(v, h)$ are welding speed and the thickness of the weld joint, respectively [13]. The predicted cooling time found regarding the samples was approximately 8.9 s. To better understand the effects of the heat input on welding, the cooling rate directed the precise prediction of the welding

that was connected to the welding parameters. Additionally, as stated by Guo, W. [10], the cooling rate had an essential role in welding through its effect on the microstructure. Thus, it is known that with the increase in the heat input, the cooling rate was slower.

In the present research, the purpose of using a computational method was to approximate the real condition through the evaluation of structural and thermal behaviors of different scenarios without the necessity of producing a real specimen. The following results present mainly different conditions of the heat input that were derived from the welding parameters. As shown in Figure 12b, it was considered single-pass welding at 0.49 kJ/mm of the heat input, which corresponded exactly to the same value on the physical specimen. In this way, the intention to use the same condition was due to the possibility of "calibrating" the software and reproducing other welding results. However, it was possible to visualize some differences between the intermediary probes' results if both the physical and the simulation were compared, which brought us to the necessity of having other methods related to machine learning to refine the results. Additionally, it was found that similarity with the peak temperature at around 550 °C was achieved on temperature probe 1 along the physical model. The cooling rate of the same probe was approximately 47 °C/s, which is closer to the 42 °C/s found on the physical specimen. In Figure 12c, the thermal cycle shown is still related to a single one. However, the welding parameters were changed to achieve a higher heat input of 0.84 kJ/mm compared to the previous test. Thus, the peak temperature of probe 1 was around 900 °C, which is a visibly elevated temperature if compared to the previous conditions. If we focused on the microstructure, the transformation of the material in the tested heat input was estimated to be predominantly bainite with a mixture of martensite, which is similar to the 0.49 kJ/s heat input. This effect could be evidenced by the cooling rate of approximately 19 °C/s, which is slower if compared to the 0.49 kJ/mm heat input used previously. Another similarity between both tested results was the estimation of a hardness result that was approximately 325 HV, which is slightly lower than the one found on the physical and virtual samples of 0.49 kJ/mm. To evaluate not only different welding parameters but also different welding conditions, Figure 12d shows the results from the thermal cycle in a simulation of multi-pass welding, where three welding passes were performed. In this type of welding condition, inter-pass welding is a factor that must be considered to evaluate the welding heat input. In the computational simulation, the nominal value of the 0.49 kJ/mm heat input used on the physical and virtual specimens was also considered in this simulation to ease the comparison. Thus, the inter-pass welding simulation presented a difference in the temperature from each welding pass at around 200 °C. In this way, due to the influence of the heat input in a single-pass welding result of 0.84 kJ/mm, we could infer that at a high heat input, the inter-pass temperature would be higher. The inter-pass control temperature is essential to avoid some defects in welding. For example, Jenney, C. [16] stated that one method to minimize intergranular corrosion on welding is to maintain an inter-pass temperature under 121 °C. Thus, due to different requirements, the inter-pass temperature had to be controlled, and by using the welding simulation, this condition might be predicted. Another consideration of multi-pass welding regards the microstructure changes that are more influenced by the sharp thermal differences in the same structure.

An overview of both single- and multi-pass welding is shown in Figure 13. The extension of the HAZ is observed along a temperature color map, where the highest temperatures were around the heat source. Based on the image of the multi-pass welding, it was clear that due to the welding passes, there was cumulative heat input that widened the extension of the HAZ. In research related to the influence of different welding inputs on S1100QL material, Mehran et al. [32] evidenced that an increase in the heat input on the joint influenced a change in the microstructure by tempering the martensite structure on the welding region. In addition, it was also observed that good control of the heat input could lead to the necessary equilibrium between the weld hardening and tempering. These methods are essential for predicting the final strength result of a determined structure.

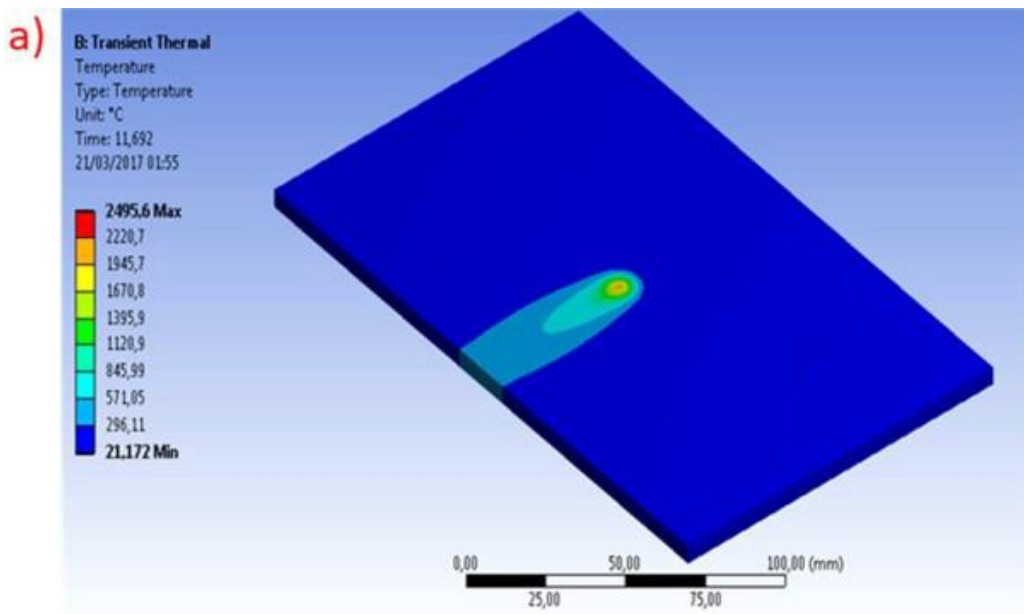

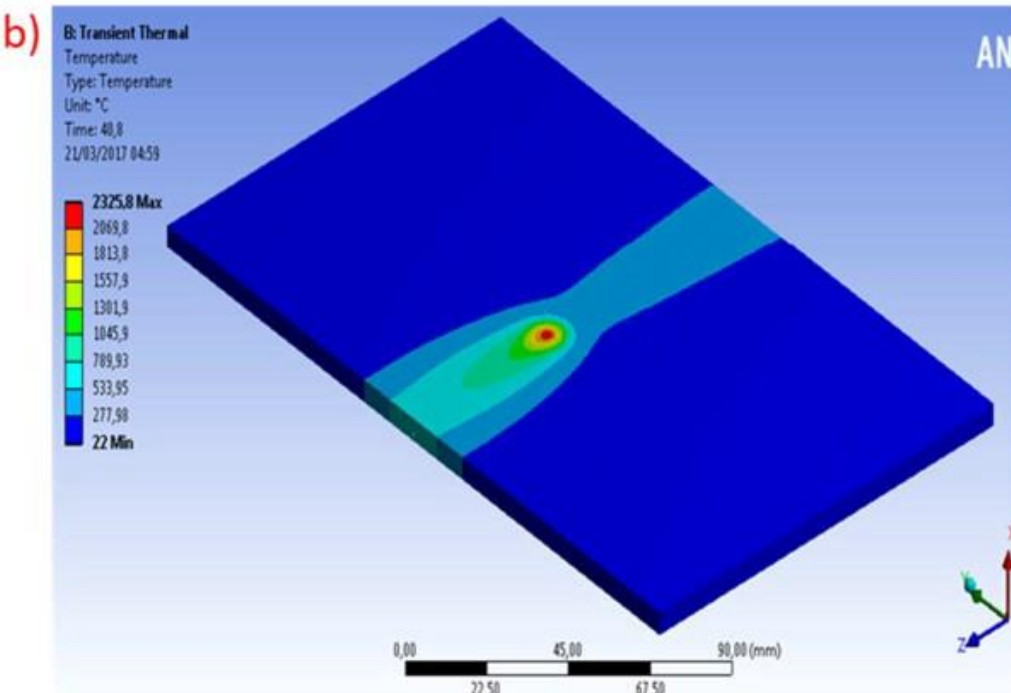

**Figure 13.** Temperature distribution on welding at (**a**) Q = 0.49 kJ/mm, v = 5 mm/s, single-pass and (**b**) Q = 0.49 kJ/mm, v = 5 mm/s, multi-pass.

The difference between the single- and multi-pass heat input was better observed when both cross-cut sections were compared. Therefore, Figure 14 shows a comparison between the temperature distribution on a cross-section of each virtual sample when both were Q = 0.49 kJ/mm and v = 5 mm/s. Accordingly, it was visible that the HAZ of the multi-pass condition was almost two times higher than the single-pass. It is important to mention that the image captured from the software could be related to the third pass performed on the virtual sample to evaluate the worst condition of the heat input from multi-pass welding.

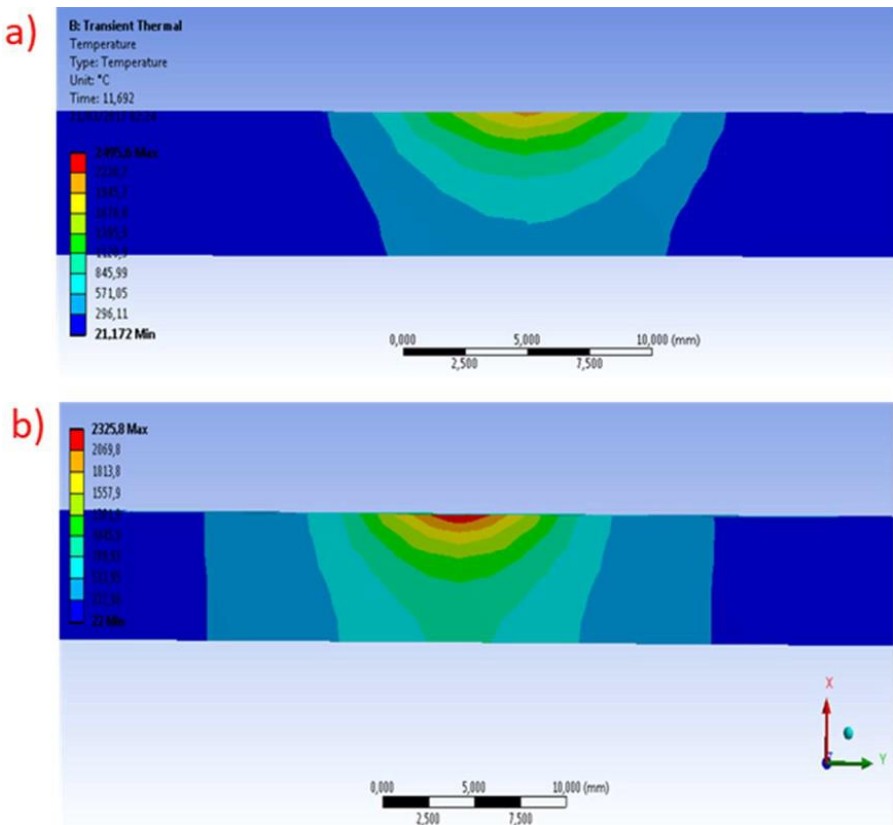

**Figure 14.** Temperature distribution on cross-section welding at (**a**) Q = 0.49 kJ/mm, v = 5 mm/s, single-pass and (**b**) Q = 0.49 kJ/mm, v = 5 mm/s, multi-pass.

To validate the results obtained by computational simulation, a transversal cut was accomplished on the virtual sample and then compared to the real specimen (Figure 15). This analysis provided the possibility of estimating the extension of HAZ based on the temperature distribution obtained. In general, as the fusion zone was closer and more centered from the top surface of the plates, where the heat source was located, the temperature was higher and more concentrated. In Figure 15, it is also possible to see that the shape created by the GMAW process was similar to the one found on the virtual model. Hence, this shape was expected, and it emphasized the good alignment between both virtual and experimental models.

The comparison of the fusion zone through an image of both models is a rough demonstration; however, with the correct software resources available, the predictions can be performed by only changing the welding parameters to the best fit of the real condition, giving the necessary assumptions of a real condition. Additionally, as stated by Goldak, J. [17] and Karkhin et al. [18], a computational simulation to predict the behavior of HAZ provides many benefits, mostly due to the elevated sensitivity of the HAZ, which is also followed by the high dependency of the microstructure behavior of that region.

Since the present study focused on heat input, it is important to relate the impact of this factor not only to the microstructure but also to the deformation along with temperature and time. The deformation was not measured on the physical specimen; therefore, it was not possible to form a comparison. The thermo-mechanical characteristics were evaluated by the same software used on the thermal analysis, though another module was used to insert the information already collected from the thermal analysis on a thermo-mechanical evaluation. The intent to present this result was to exemplify another feature that is available to aid in predicting the behavior of a certain welding parameter. Thus, in Figure 16, the equivalent stress shows the critical regions and the stress peak of the sample. This type of analysis is essential during the development of a welded structure since it

allows the evaluation of larger assemblies where many parts are involved, and a stressed region could influence the failure of the whole structure.

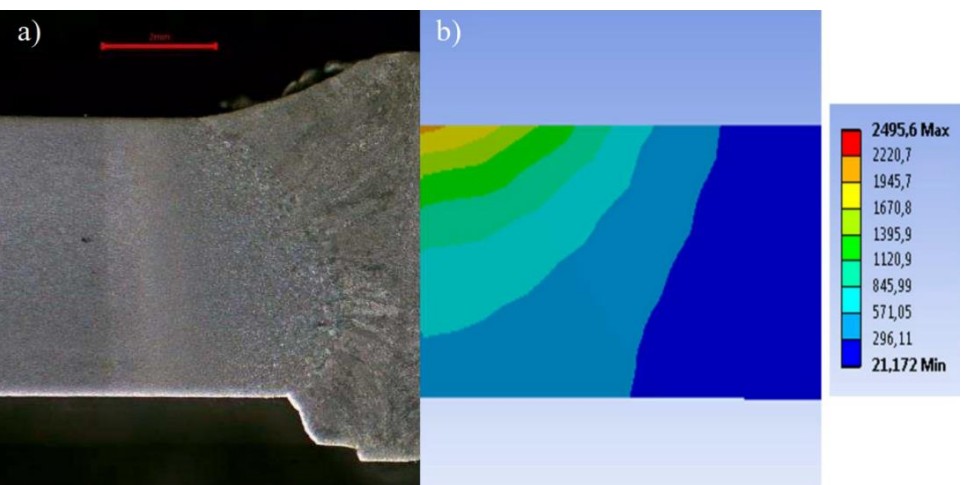

**Figure 15.** Comparison of the fusion zone between the physical (**a**) and the virtual (**b**) experiment.

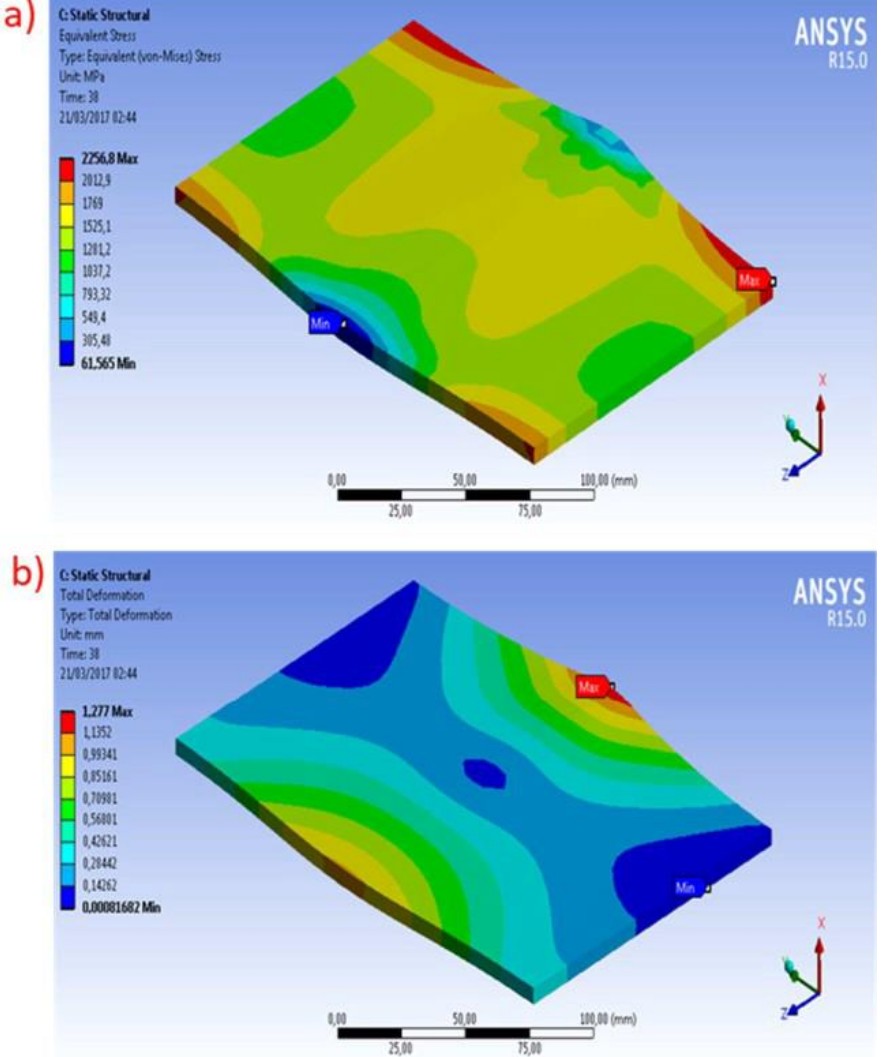

**Figure 16.** Equivalent stress (**a**) and total deformation (**b**) on welding at Q = 0.49 kJ/mm, v = 5 mm/s, single-pass.

In addition, the total deformation is also presented in Figure 16, presenting the variation in millimeters of the single-pass condition of welding with Q = 0.49 kJ/mm of the heat input. The results from the computational method presented a 1.277 mm deformation in some specific regions. Once more, this welding simulation feature promoted a better design for the parts or weld assembly to ensure that all the aspects influenced by the heat input were covered during the development.

## 4. Conclusions

This research's most important and notable finding is the effect of heat input on UHSSs and the different methods that could be used to predict its impact. By considering a physical experiment or a computational simulation, it was possible to prove that some welding parameters, like current, voltage, and welding speed, directly impact the heat input. Furthermore, based on the results found, the GMAW process was suitable for obtaining great results with welding UHSSs. However, due to the elevated heat input that this welding process can achieve, it is important to consider welding features that might control or provide a lower heat input, such as laser or pulse welding.

The prediction of the behavior of welding could be accomplished using the computational welding simulation. Through the use of FEA, it was possible to estimate the extension of the HAZ and then predict the result of the welding and its impact on the microstructure. The cooling time is a variable that could be predicted and noticed as the advantage of considering a range of 15 s on UHSS materials. Additionally, for obtaining high-quality and reliable welding, the weld pool size and shape had a considerable influence in terms of achieving a satisfactory result. Thus, the welding joint also requires special consideration since the lower the volume that is melted, the lower the heat. The recommended heat input varies according to the thickness of the material; nevertheless, in this study, it was shown that a heat input of 0.49 kJ/mm was appropriate for the 5 mm thick plate.

Another important result was the influence of the cumulative heat input on the joints through multi-pass welding, which greatly affected the microstructure of UHSSs as compared to a single-pass welding joint. In this way, instead of improving only the welding parameters, an alternative was to change the design of the structure to avoid multi-passes. It was also found that UHSS is an alternative solution to reducing the thickness of structures through its strength benefits, which consequently can reduce the number of welding passes. Accordingly, the peak temperatures initially obtained could be used as an input to evaluate the thermo-mechanical effects. With an increase in the heat input of 8.4 kJ/cm, the estimated cooling rate was around 70 °C/s. There was the presence of a softening area in the coarse grain heat-affected zone (CGHAZ) of the welded joints. These results led to an increase in the carbon content (3.4%) compared to the base metal around the fusion zone of the weld joints.

Finally, the utilization of UHSS, considering its benefits to a welded structure, has been demonstrated to be a suitable choice because it offers great versatility for use in different applications. In addition, this new generation has shown obvious improvements, and this research could provide evidence that it is possible to achieve excellent results for welded structures made with UHSSs. However, a challenge in welding UHSSs consists of the several ranges of material and grades with different heat treatments and alloys, which could bring difficulties to the prediction method. However, the spread of machine learning along with real-time monitoring has the potential to bring features that may ease the welding predictions of UHSSs. Moreover, as with most of the prediction methods, the computational welding simulation approximates a real condition and must be faced as a tool that could never replace reality but assist decision-making.

**Author Contributions:** Conceptualization, F.M.N.B. and P.K.; methodology, A.N.; software, A.N.; validation, F.M.N.B., P.K. and A.N.; formal analysis, F.M.N.B.; investigation, A.N.; resources, P.K.; data curation, P.K.; writing—original draft preparation, F.M.N.B.; writing—review and editing, P.K.; visualization, F.M.N.B.; supervision, P.K. All authors have read and agreed to the published version of the manuscript.

**Funding:** This research received no external funding.

**Conflicts of Interest:** The authors declare no conflict of interest.

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
