# Peer review of "Optimization of GMAW Process Parameters in Ultra-High-Strength Steel Based on Prediction"

_metals, doi:10.3390/met13081447_

Round 1

Reviewer 1 Report

1. The effect of heat input on the microstructure of UHSSs  in the multi-pass welding and single-pass welding joints should be concluded clearly.

2. The advantage of the computational analysis is based on the interactivity of different interfaces that may predict not only the heat input effect but also the analysis of the welding by its distortion and shrinkage consequences. This conclusion has not been supported by the results.

Author Response

Dear Reviewer,

Thank you very much for you comments and suggestions.

Please, find below some results

  1. The effect of heat input on the microstructure of UHSSs in the multi-pass welding and single-pass welding joints should be concluded clearly.

In the analysis, the results presented the fact that an increase of heat input of 8.4kJ/cm, the estimated cooling rate was around 70 ºC/s. they were a presence of softening area in the coarse grain heat affected zone (CGHAZ) of welded joints. This results lead to the increase of carbon content (3.4%) compare to the base metal. SEM-EDS mapping were done only when the heat input was 8.4 kJ/cm.  

  1. The advantage of the computational analysis is based on the interactivity of different interfaces that may predict not only the heat input effect but also the analysis of the welding by its distortion and shrinkage consequences. This conclusion has not been supported by the results.

Thank you for the comments.

In this analysis, the distortion and shrinkage consequences were not evaluated.

The area where replace by the effect of cooling in the microstructure constituent and the softening area of the HAZ.

Based on what the analysis was focused, it was noticed the fact that the UHSS is an alternative solution to reduce the thickness of structures through its strength benefits, which consequently may reduce the number of welding passes. Besides, another advantage of the computational analysis is based on the interactivity of different interfaces that may predict not only the heat input effect but also the analysis of the welding by its distortion and shrinkage consequences. Accordingly, the peak temperatures initially obtained may be used as an input to evaluate the thermo-mechanical effects.

Reviewer 2 Report

The authors investigated Optimisation of GMAW Process Parameters in Ultra-high Strength Steels Based on Prediction.The investigation is not innovative enough and the experimental results doesnt match the research objectives.

1. The GMAW process was suitable can be valuated by simpler method for example,WPQ.

2. I didnt see the relationship between the finite element mode result and the microstructure,mechanical properties.So you should provide the detailed process parameters and how it relates to the microstructure,mechanical properties.

3.The microstructure and Micro-hardness of the symmetrical location of the five thermocouples probes could be offered.The relationship between heat input and microstructure, mechanical properties would be more clear.

  • There are many grammar problems in the paper,English language should be further improved.

  •  

Author Response

Thank you very much for the comments and suggestions,

Please, find below les answers to the comments and suggestions

The authors investigated Optimisation of GMAW Process Parameters in Ultra-high Strength Steels Based on Prediction. The investigation is not innovative enough and the experimental results doesn’t match the research objectives.

  1. The GMAW process was suitable can be evaluated by simpler method for example,WPQ.

Thanks,

In this research, it was also about to use a numerical and experimental model to evaluate a cooling rate and experimentally how the cooling process can affect microstructure specially where there is an increase of softening.

  1. I didn’t see the relationship between the finite element mode result and the microstructure, mechanical properties. So you should provide the detailed process parameters and how it relates to the microstructure, mechanical properties.

Thank for the comment.

They were experimental mechanical tests in this analysis.

Only a thermal analysis, micro hardness, and microstructure behaviour that were found in the experimental procedure. They were a correlation in between a cooling rate end microstructural constituent in different areas of the welded joints.

It found that,

An increase of heat input of 8.4kJ/cm, the estimated cooling rate was around 70 ºC/s. they were a presence of softening area in the coarse grain heat affected zone (CGHAZ) of welded joints. This results lead to the increase of carbon content (3.4%) compare to the base metal, around the fusion zone of the weld joints. 

  1. The microstructure and Micro-hardness of the symmetrical location of the five thermocouples probes could be offered. The relationship between heat input and microstructure, mechanical properties would be clearer.

Thank you.

From the location of five thermocouples probes in the finite element model and an experimental model, only the location 1 which was close to the weld metal was analysed in point of view microstructural constituent.

Comments on the Quality of English Language

  • There are many grammar problems in the paper English language should be further improved.

Thank you. The authors have made some improvements in the manuscript.

Reviewer 3 Report

This manuscript investigated the best parameters for welding the S960 material based on prediction methods. The heat input responsible for the effect of UHSS microstructure was discussed. Finite element analysis (FEA) was used to simulate and evaluate the result, and some interesting results were got. However, some items should be well clarified.

1.       Abbreviations should be given their full name the first time they appear, such as “GMAW”.

2.       The yield strength of ultra-high strength steel is generally greater than 1180MPa, and the ultimate tensile strength is greater than 1380MPa. The mechanical properties of metal UHSS S960 is lower than this. Can it be called ultra-high strength steel?

3.       Line 142, Equation 1 may be wrong, please check the criteria again.

4.       Line 333, which etchant was used to etch cross-section?

5.       Line 414, dose Figure 11a represent fusion zone (FZ)? From the manuscript, the HAZ has a large area.

6.        Line 452, “Δt8-5 is the difference……” while in equation (4) is Δt800-500. Are these two ways of writing the same?

7.       The English should be polished by a native speaker. Too many bad expression and non-professional words exists in the manuscript.

The English should be polished by a native speaker. Too many bad expression and non-professional words exists in the manuscript.

Author Response

Comments and Suggestions for Authors

Dear reviewer,

Thank you very much for your comments and suggestions.

Please, find below some answers to the comments and suggestions.

This manuscript investigated the best parameters for welding the S960 material based on prediction methods. The heat input responsible for the effect of UHSS microstructure was discussed. Finite element analysis (FEA) was used to simulate and evaluate the result, and some interesting results were got. However, some items should be well clarified.

  1. Abbreviations should be given their full name the first time they appear, such as “GMAW”.

Thank you very must.

The full name has been added in the manuscript (abstract and in the introduction)

  1. The yield strength of ultra-high strength steel is generally greater than 1180MPa, and the ultimate tensile strength is greater than 1380MPa. The mechanical properties of metal UHSS S960 is lower than this. Can it be called ultra-high strength steel?

Thank you for the comment.

There is a different classifications of modern steel.

At this time it can be site

Mild steel

High strength steels (HSS)

Very high strength steel (VHSS)

Advanced high strength steel (AHSS)

Ultra high strength steel (UHSS)

There is several definition regarding UHSS. Based on the perspective of material strength:

  1. The yield strength of the material is over 560 MPa;
  2. The tensile strength of the material is over 700 MPa;
  3. The yield strength of the material is 900 MPa and Above;
  4. The tensile strength of UHSS is up to 1700 MPa, especially

for the martensitic steels

  1. Line 142, Equation 1 may be wrong, please check the criteria again.

Thank you the equation have been corrected.

  1. Line 333, which etchant was used to etch cross-section?

The sample specimen was etching at the initial solution of 4% solution of HNO3 in ethanol [39].

The same solution was used to carry out a microstructure analysis.

  1. Line 414, dose Figure 11a represent fusion zone (FZ)? From the manuscript, the HAZ has a large area.

Thank you for the comment.

Figure 11a in the figure show the weld metal microstructural constituent, which is  mainly the presence of acicular ferrite (AF) and a consequence of the amount of pro-eutectoid ferrite (PF) at the temperature of around 600 0C

  1. Line 452, “Δt8-5 is the difference……” while in equation (4) is Δt800-500. Are these two ways of writing the same?

Thank you very much.

The question has been review and a new cooling rate at 500 oC is shown in the manuscript

The cooling rate value at 500 0C (R500) was based on the following equation  

                                                              (4)

In Equation 4,  is defined as the increase of the temperature during welding,  the heat density.  the thermal conductivity, weding speed and the thickness of the weld joint

  1. The English should be polished by a native speaker. Too many bad expression and non-professional words exists in the manuscript.

Comments on the Quality of English Language

The English should be polished by a native speaker. Too many bad expression and non-professional words exists in the manuscript.

Thanks.

There are some improvement regarding the English check of the manuscript.

Reviewer 4 Report

The manuscript is written well and it can be considered after following modifications

1) Source of tables should be added.

2) Table 7 is not readable.

3) You added in the last paragraph of introduction that a comparison table is added. But that comparison table is missing. A comparison to previous work should also be added.

4) Description of method should be added more.

Moderate English changes are required.

Author Response

Thank you very much for your comments and suggestions regarding the manuscript.

Please, find below the answers regarding the manuscript.

The manuscript is written well and it can be considered after following modifications

  • Source of tables should be added.

The source of tables are now available in the manuscript (page 4)

2) Table 7 is not readable.

Thank you

Table 7 has been added in the manuscript

Table 7.  Welding parameters used during the welding simulation by finite element.

Welding passes

Voltage (V)

Current (A)

Welding speed (mm/s)

Efficiency (%)

Heat input (kJ/mm)

1

24

180

7

80

0.49

2

30

290

8.3

80

0.84

  • You added in the last paragraph of introduction that a comparison table is added. But that comparison table is missing. A comparison to previous work should also be added.

Thank you. The comparison table is added now.

  • Description of method should be added more.

A hardness test was conducted for a line of measurement using a Wilson Wolpert 452SVD Vickers hardness tester (ITW, Chicago, IL, USA) according to ISO 6507-1:2018. The measurements were taken point-by-point over the entire length of the sample. Figure 4 shows the scanning electron microscopy (SEM) and energy-dispersive X-ray spectroscopy (EDS) device (Hitachi SU3500, America, Chicago, IL, USA). EDS identified the variation of the alloy element compositions in the WM, CGHAZ, and FGHAZ. To carry out those tests, the sample was first of us have to be dipped in acetone solution to remove the accumulated carbon layer from the sample surface. The liquid must be removed with a CO2-skimming gun to obtain a dry surface. After that, the sample was placed inside the SEM/EDS device for checking with different magnifications.

Comments on the Quality of English Language

Moderate English changes are required.

Thank you.

English language have been checked.

Round 2

Reviewer 2 Report

The revised manuscript had provided more sufficient reason.

Does different the locations of five thermocouples probes present different view microstructural constituent? Does there the same regularity exist like the location 1? Please explain the rationality of choose one location of thermocouples probes.

Author Response

Thank you very much for the comments and suggestions.

Does different the locations of five thermocouples probes present different view microstructural constituent? Does there the same regularity exist like the location 1? Please explain the rationality of choice one location of thermocouples probes.

For the microstructure analysis, the choice was based on the result of softening area during hardness analysis.

Reviewer 3 Report

Please check the fomular 1, again.

https://hha.hitachi-hightech.com/en/blogs-events/blogs/2020/10/26/carbon-equivalent-fundamentals-for-predicting-steel-properties/

The English is OK.

Author Response

Dear Reviewer,

Thank you very much for the comment and recommendations.

Formulation 1 has been corrected

Reviewer 4 Report

The revised manuscript is quite better than the original.

Moderate English changes like spell check is required.

Author Response

Thank you very much for your comments and suggestions.

The English language has been checked.

I think now it is ok